# Predictors of Mental Health Outcomes in a Multidisciplinary Weight Management Program for Class 3 Obesity

**DOI:** 10.3390/nu16071068

**Published:** 2024-04-05

**Authors:** Ashley Lam, Milan K. Piya, Nasim Foroughi, Mohammed Mohsin, Ritesh Chimoriya, Nic Kormas, Janet Conti, Phillipa Hay

**Affiliations:** 1School of Psychology, Western Sydney University, Sydney, NSW 2560, Australia; j.conti@westernsydney.edu.au; 2Translational Health Research Institute, School of Medicine, Western Sydney University, Campbelltown, NSW 2560, Australia; m.piya@westernsydney.edu.au (M.K.P.); n.foroughi@westernsydney.edu.au (N.F.); ritesh.chimoriya@health.nsw.gov.au (R.C.); 3Camden and Campbelltown Hospitals, Campbelltown, NSW 2560, Australia; nic.kormas@health.nsw.gov.au; 4Mental Health Research and Teaching Unit, Liverpool Hospital, South Western Sydney Local Health District, NSW Health, Liverpool, NSW 1871, Australia; m.mohsin@unsw.edu.au; 5Faculty of Medicine & Health, School of Clinical Medicine, Discipline of Psychiatry, University of New South Wales, Kensington, NSW 2052, Australia

**Keywords:** obesity, class 3 obesity, psychological distress, quality of life, eating disorders, weight management, mental health

## Abstract

This study aimed to examine the potential predictors of improvement in mental health outcomes following participation in an intensive non-surgical outpatient weight management program (WMP) in an Australian public hospital. This was a retrospective cohort study of all adults with Class 3 obesity (BMI ≥ 40 kg/m^2^) who enrolled in the WMP from March 2018 to June 2021. The participants completed the Eating Disorder Examination Questionnaire Short Version (EDE-QS), Kessler-10 Psychological Distress Scale, and 36-Item Short-Form Survey (SF-36) at baseline and 12-month follow-up. A total of 115 patients completed 12 months in the WMP and were included in the study, with 76.5% being female, a mean ± SD age at baseline of 51.3 ± 13.8 years, a weight of 146 ± 26 kg, and a BMI of 51.1 ± 8.6 kg/m^2^. The participants lost an average of 8.6 ± 0.2 kg over 12 months, and greater weight loss at follow-up was significantly associated with improved global EDE-QS scores, psychological distress, and improved mental health quality of life. However, improvements in most mental health outcomes were not predicted by weight loss alone. Notably, a lower eating disorder risk at baseline was associated with less psychological distress at follow-up and greater weight loss at follow-up. Our results also found an association between reduced psychological distress and reduced binge eating frequency. These findings support the inclusion components of obesity interventions that target the psychological correlates of obesity to support improved outcomes in people with Class 3 obesity. Future studies should aim to identify which aspects of the WMP helped improve people’s psychological outcomes.

## 1. Introduction

Obesity is a complex metabolic disease affecting 13% of the global adult population, tripling in prevalence from 4% in 1975 to over 18% in 2016 to become an epidemic in developing and developed countries [1]. The negative health impacts of obesity follow a graded curve indicating the severity of obesity, such that individuals with obesity Classes 2 and 3 (BMI from 35 to 40 and BMI > 40 kg/m^2^, respectively) are at a higher risk of adverse consequences and a higher severity of comorbidities [2,3]. As a result, Class 2 and 3 obesities in adults are associated with an increased risk of disability and reduced life expectancy with an associated significant economic burden in terms of healthcare cost and the overall cost to society [4,5].

Obesity has also been consistently associated with adverse mental health issues, such as depression, anxiety and eating disorders [6,7,8,9], and reduced quality of life (QoL) even in the absence of other chronic health conditions [10,11,12]. The prevalence of psychological disorders has been found to be significantly higher in individuals with obesity than in their healthy comparators, and the presence of both confers a lower quality of life and increased healthcare use than either condition independently [13,14]. Furthermore, mental health challenges, particularly serious mental health illnesses, increase the risk and parallel the severity of obesity [3]. Additionally, individuals with more severe obesity, and consequently more comorbidities, have been found to experience more social disadvantages, thereby increasing the risk of worsening mental health [15,16,17]. Consequently, researchers have emphasised the need for effective, accessible, and affordable weight management interventions to address this bidirectional and complex inter-relationship between the psychological and physical correlates of obesity [18,19].

The psychological outcomes of weight management programs (WMPs) have primarily focused on changes in depression, anxiety, and health-related quality of life. Improvements in mood have been found in a number of randomised controlled trials following WMPs compared to active (intervention) [20,21,22,23,24] or non-active (waitlist or treatment as usual) comparator groups [20,25,26]. While some studies reported no significant improvement in depression in the short term, these could be attributable to the online mode of delivery [27,28] due to the nature of the study design and insufficient treatment doses [28,29]. Longer-term outcomes (greater than 12 months post-treatment) in most of the studies have found sustained improvements in depressive symptoms [30,31,32,33], although the superiority of the intervention group is not always clear. This may be explained by the hypothesis that the effects of the intervention may only be evident if weight loss had been maintained [34,35]. Nevertheless, weight regain in the long term was not found to be associated with a lapse into depressive symptoms [35].

Several studies have also found participants experienced reduced anxiety compared to those at the baseline [26] when treated in WMPs compared to a control group [25]. One study, however, by Heath and colleagues [24] found no evidence of improvement from baseline and non-significant differences between the intervention and control groups. However, this may be due to ceiling effects, given that the mean baseline anxiety scores fell in the normal range. In the long term, other studies generally found no changes in anxiety at 12 and 24 months and no differences between the control groups [31,34,35]. Overall, recent studies have demonstrated a small positive effect on anxiety in the short term that are not maintained in the long term.

Several recent studies have also reported improvements to varying degrees with WMPs, with some finding short-term improvements in both mental health quality of life (MHQoL) and physical health quality of life (PHQoL) [36], while others found improvements in only MHQoL [37]. However, this may be explained by the study population, for example, pain may have contributed to the participants’ ratings of PHQoL [37]. In the long term, the findings have been inconsistent, with one study reported improvements in PHQoL, but not MHQoL [38]; another study found the opposite [32], and others indicated no differences between the intervention and control groups in short or long term [27,29,34]. This inconsistency in findings may reflect the close association of MHQoL with weight change, even with the use of a more general measure. Improvements in QoL appear to be largely affected by weight change, particularly the physical component [36,37,38]. Altogether, the findings suggest behavioural interventions produce improvements in QoL contingent on improvements in weight reduction, where this relationship is more pronounced for PHQoL.

Although WMPs appear to have a general positive impact on mental health, not all individuals receive equal benefits. For example, Young and colleagues [26] found that while the WMPs had positive effects on depressive symptoms, the effect was greater for men who had a higher BMI along with greater access to mental health treatment. This emphasises that behavioural interventions do not impact all the participants’ mental health and wellbeing in equivalent ways. The differences in treatment efficacy may be partly accounted for by the participants’ baseline characteristics, such as starting levels of depression and anxiety, starting levels of eating disorder symptomatology, and starting BMI. Although BMI is the most commonly used measure in WMP efficacy studies, this presents as a limitation given the potential for the misclassification of obesity risk [39]. In recent years, several limitations of BMI measures have been reported, e.g., it cannot distinguish the changes in body composition [40]; it does not account for body fat distribution [41]; and it does not consider age, gender and ethnicity in its estimation [42,43,44]. While other studies have indicated that the Dual-energy X-ray absorptiometry (DEXA) technique can be used as the gold standard, which gives a reasonably more accurate body composition estimation than BMI [45,46], this may not always be feasible in people with a very high weight. Despite these limitations, due to the simplicity and convenience of data collection, this study used BMI as a measure of obesity. Previous qualitative research has also suggested that mental health, stigma, and social isolation impact engagement with community-based services [47]. While the baseline QoL scores have been used to try and predict weight loss after bariatric surgery [48], to the author’s knowledge, there is no current research on the psychological predictors of mental health outcomes following non-surgical weight management interventions. Most research has focused on identifying the predictors of mental health outcomes after bariatric surgery, although there are not many studies in this area [49].

Overall, multidisciplinary WMPs generally seem to produce significant improvements in depression that are maintained long term, with short-term improvements in anxiety. The improvements in QoL may be dependent on the weight loss outcomes and may be influenced by the pre-existing mental and physical health co-morbidities, particularly for PHQoL. However, there is a paucity of research investigating participants’ characteristics (such as psychological distress, eating disorder symptomatology, QoL and BMI) that may predict better responses to treatment. Therefore, this study examines the impact of the participants’ baseline BMI as well as clinical and sociodemographic characteristics on improvement in mental health outcomes among people with Class 3 obesity following WMP intervention, controlling for age, gender, and weight loss.

## 2. Materials and Methods

### 2.1. Study Design and Participants

This study used a retrospective cohort design and was conducted in a publicly funded hospital-based outpatient program in Sydney, Australia. All the participants were above the age of 18 and categorised as having Class 3 obesity (BMI ≥ 40 kg/m^2^) with at least one medical comorbidity related to their weight and were enrolled between March 2018 and June 2021 in this multidisciplinary outpatient WMP. The WMP offers clinical obesity services to residents in South Western Sydney, which represents some of the most socioeconomically disadvantaged communities in New South Wales, Australia. Referrals to the WMP could be made by any medical practitioner, and the multidisciplinary team included endocrinologists, dietitians, psychologists, physiotherapists, a gastroenterologist, a psychiatrist, and a specialist nurse. Each component of the WMP and the cohort study has been previously published [15,50,51,52,53]. The team provided care individually tailored to each patient, which included one or more of the following: lifestyle interventions, behavioural modifications, dietary interventions, and pharmacotherapy. The use of pharmacotherapy was very limited within the program, as the cost was not subsidised by the Federal Government Pharmaceutical Benefits Scheme in Australia, and the patients in the publicly funded program were unable to afford the cost of private prescriptions. While the people in the program were eligible for bariatric surgery, none of the patients in this cohort underwent surgery within the first 12 months.

All the enrolled patients were seen by a physician and members of the multidisciplinary team, including a psychologist and dietitian. Patients assessed as having a high risk of eating disorders or mental health issues were then referred to the psychiatrist. During the study period, a total of 306 eligible participants were enrolled in the WMP. Prior to program intervention, all the enrolled participants were assessed through a self-repot baseline questionnaire, and the participants were re-assessed at the 12-month post-assessment follow-up. Of the 306 enrolled participants, only 115 (37.6%) completed both the baseline and 12-month post follow-up assessments, and the remaining 191 (62.4%) did not complete the 12-month post-assessment follow-up. One hundred and fifteen participants who completed assessment at both time points were included in this study.

### 2.2. Data Collection and Management

Data on weight, height, medical comorbidities, current medications, and blood tests were collected directly during initial presentations to the clinic at baseline and at 12-month follow-up if they were still attending the program. At these time points, all the participants were also requested to complete three self-administered questionnaires, Eating Disorder Examination Questionnaire Short Version (EDE-QS), Kessler-10 Psychological Distress Scale (K10), and 36-Item Short-Form Survey (SF-36), each of which will be discussed in further detail below. The data were anonymised data that were collected routinely in the clinic, and a waiver for signed informed consent was granted as part of a quality improvement project.

#### 2.2.1. Risk of Eating Disorders (EDE-QS) and Binge Eating Frequency

The EDE-QS was administered to assess the frequency or severity of eating disorders symptoms for anorexia nervosa, bulimia nervosa, and binge eating disorder over the past seven days [54]. The patients responded to each of the 12 items on a four-point rating scale ranging from 0 (“0 days” or “not at all”) to 3 (“6–7 days” or “markedly”). The scores were then summed to obtain a total score between 0 and 36, where higher scores indicate more frequent and severe eating disorder symptomatology, and therefore a higher risk of eating disorder [54,55]. A total score of 15 has been proposed as the threshold score to identify a high risk of developing an eating disorder [54]. In this study, Cronbach’s alpha (α) for the item pool of EDE-QS Global score was 0.80 at baseline and 0.82 at the 12-month follow-up assessment.

The EDE-QS item 10 ‘On how many days on which you had a sense of having lost control over your eating did you eat what other people would regard as an unusually large amount of food in one go?’ was used to assess the quantity of binge-eating episodes during the last 7 days. Responses were scored on a four-point rating scale, ranging from 0 days to 6–7 days.

#### 2.2.2. Psychological Distress (K10)

The K10 was administered to assess non-specific psychological distress. The participants indicate the severity and frequency of negative mood, anxiety, and emotional states over the last four weeks, providing ratings on a five-point Likert scale, ranging from 1 (“none of the time”) to 5 (“all of the time”) [56]. The scores were then summed to obtain a total score between 10 and 50, where higher scores indicate higher levels of psychological distress [57]. The total score indicates the levels of psychological distress as follows: low (10–15), moderate (16–21), high (22–29), and very high (30–50) [58]. In this sample, Cronbach’s alpha (α) for the item pool of K10 score was 0.93 at baseline and 0.94 at 12-month follow-up.

#### 2.2.3. Health-Related Quality of Life (SF-36)

The Short-Form Survey (SF-36) was administered to assess health-related QoL. It consists of eight subscales measuring physical functioning, role limitations due to physical factors, role limitations due to emotional factors, vitality, mental health, social functioning, bodily pain, and general health [59]. Subscales can be aggregated into two component summary scales that range from 0 to 100; the Physical Component Summary (PCS) provides a measure of physical health, while the Mental Component Summary (MCS) provides a measure of mental health [60]. Higher scores on the PCS and MCS indicate a better QoL. The SF-36 has shown good internal consistency in patients with obesity, with a Cronbach’s alpha of 0.717 [61].

Scores for the eight subscales were calculated based on the responses to the items on each scale using standard SF-36 scoring algorithms [59]. The scores for each subscale were then standardised using z-score transformation and SF-36 scale means and standard deviations for Australian adult population norms [60,62]. Following this, the standardised scores of the eight subscales were summed to calculate the PCS and MCS scores.

### 2.3. Statistical Analysis

Descriptive statistics of baseline characteristics were calculated, and mean total scores with 95% confidence intervals (CIs) are presented for all the outcome measures, stratified by gender at the baseline and 12-month follow-up. The overall mean total scores for all the outcome measures were compared between genders across each time point. To examine the pairwise significant differences between the baseline and 12-month follow-up, a paired *t*-test was used for all the outcome measures, controlling for gender. Pearson’s correlation coefficient was also computed for each outcome measure to measure the strength of the relationship between two time points, controlling for gender. Based on two-way comparisons, effect sizes (Cohen’s *d*) were computed for each outcome as an indication of the magnitude of change in each outcome from baseline to follow-up [63]. Initial analysis found that there was no statistical difference in the psychological outcome measures when comparing the different employment status groups. Also, the sample sizes for different employment groups were not statistically sufficient to explore the effect size for each of the employment groups, so employment status was excluded from statistical analyses.

Finally, associations between the outcome measures and patients’ characteristics were examined. Measures found to be statistically significant with each outcome were included in multiple linear regression analysis to examine the relative contribution of each predictor, controlling for age, gender, and weight loss. To avoid multicollinearity in case of correlations between the same baseline and follow-up measures, only significant follow-up measures were included in the follow-up model. All analyses were conducted by using the Statistical Package for Social Sciences, Version 28 (SPSS for Windows, IBM Corp., Armonk, NY, USA).

## 3. Results

### 3.1. Participants’ Characteristics

The baseline characteristics of the participants who completed both baseline and 12-month follow-up assessments (*n* = 115) are reported in Table 1. The participants had a mean age of 51.3 + 13.8 years at the initial assessment. Most participants were women (76.5%), more than three-quarters (78.3%) were Caucasian, and 7.8% were of Aboriginal and Torres Strait Islander descent. At baseline, the participants had a mean BMI of 51.1 ± 8.6 kg/m^2^. The clinical features reported by the sample included high levels of psychological distress (mean K10 score: 25.9 ± 9.8), a high risk of eating disorders (mean EDE-QS score: 15.9 ± 6.7), with 48% of the sample reporting episodes of binge eating at least 1 day a week, and impaired quality of life (mean PCS score: 29.5 ± 10.2; mean MCS score: 40.1 ± 12.4). The mean weight loss at 12 months in this WMP was 8.6 ± 0.2 kg (6.1%) (Table 2).

### 3.2. Outcome Measures between Baseline and Follow-Up Assessments

Table 2 reports the mean scores with 95% CIs for all the outcome measures categorised by gender for the participants who completed both the assessments. For BMI and body weight, there was a statistically significant difference between the two time points, with this effect being greater for males than that for females. In terms of psychological distress, there was a statistically significant difference between the two time points, with a similar medium effect size across genders. For the EDE-QS global score, there was a statistically significant difference between the two time points, with a similar small effect size across genders. Regarding the binge eating frequency, there was a statistically significant difference between the two time points for females, but not males, with this difference being small. For PHQoL and MHQoL, there were statistically significant differences between the two time points for both the measures. PHQoL had a medium-sized improvement for both the genders, while MHQoL had a medium-sized improvement for males and a small improvement for females.

### 3.3. Correlations among Outcome Measures

Table 3 presents a correlation matrix in which each measure is correlated with the participants’ characteristics (age, gender, and ethnicity) for both the time points (baseline and follow-up). Being female was significantly associated with having a higher BMI (r = −0.20, *p* < 0.05), greater psychological distress (r = −0.22, *p* < 0.05), and lower MHQoL (r = 0.23, *p* < 0.05). A younger age was significantly associated with higher global EDE-QS scores (r = −0.27, *p* < 0.01), more frequent binge eating (r = −0.32, *p* < 0.01), higher levels of psychological distress (r = −0.30, *p* < 0.01), higher PHQoL (r = −0.28, *p* < 0.01), and lower MHQoL (r = 0.24, *p* < 0.05). Being of Aboriginal descent was significantly associated with having a higher BMI (r = 0.24, *p* < 0.05) and higher MHQoL (r = −0.28, *p* < 0.01).

All the baseline outcome measures were found to be significantly positively associated with the same measures at 12-month follow-up. The largest correlation observed was between the baseline and follow-up BMI (r = 0.92, *p* < 0.01). A greater amount of weight loss at follow-up was significantly associated with improved global EDE-QS scores (r = −0.45, *p* < 0.01) and psychological distress (r = −0.28, *p* < 0.01) at follow-up and significantly associated with improved MHQoL (r = 0.37, *p* < 0.01).

### 3.4. Results from Multiple Linear Regression Analysis

#### 3.4.1. EDE-QS Global Score

The baseline regression models revealed that after adjustment, less psychological distress was significantly associated with a lower EDE-QS score (β = 0.63, *p* < 0.01). The regression model for the follow-up found that after adjustment, lower EDE-QS follow-up scores were associated with lower baseline EDE-QS scores (β = 0.23, *p* < 0.01), less follow-up psychological distress (β = 0.32, *p* < 0.05), and more weight loss at follow-up (β = −0.35, *p* < 0.01). The overall models for baseline and follow-up were both significant, accounting for 42.5% and 41.7% of the total variance, respectively (Table 4).

#### 3.4.2. Binge Eating Frequency

The baseline regression models showed that after adjustment, a lower binge eating frequency was significantly associated with older age (β = −0.20, *p* < 0.05) and less baseline psychological distress (β = 0.42, *p* < 0.01). The regression model of the follow-up revealed that after adjustment, a lower binge eating frequency at follow-up was significantly associated with a lower binge eating frequency at baseline (β = 0.23, *p* < 0.01) and less psychological distress at follow-up (β = 0.39, *p* < 0.01). Both the models were significant, accounting for 25.6% of the total variance at baseline and 23.7% of the total variance at follow-up (Table 4).

#### 3.4.3. Psychological Distress

The baseline regression models revealed having a lower baseline EDEQ global score (β = 0.58, *p* < 0.01) was significantly associated with less psychological distress at baseline after adjustment. The regression model of the follow-up showed that less psychological distress at follow-up was significantly associated with less psychological distress at baseline (β = 0.58, *p* < 0.01) and a lower follow-up EDE-QS global score (β = 0.31, *p* < 0.01). Both the models were significant, accounting for 46.2% of the total baseline variance and 58.3% of the total follow-up variance (Table 4).

#### 3.4.4. Physical Health Quality of Life (PHQoL)

The regression models of baseline measures revealed that a higher baseline PHQoL was significantly associated with a lower baseline BMI (β = −0.30, *p* < 0.01). The follow-up regression model revealed that a higher PHQoL at follow-up was significantly associated only with a higher baseline PHQoL (β = 0.49, *p* < 0.01). Both the baseline and follow-up models were significant, accounting for 14.7% and 29.4% of the total variance, respectively (Table 4).

#### 3.4.5. Mental Health Quality of Life (MHQoL)

The baseline regression model showed that less baseline psychological distress (β = −0.82, *p* < 0.01), a higher baseline binge eating frequency (β = 0.17, *p* < 0.01), and a lower PHQoL at baseline (β = −0.30, *p* < 0.01) were significantly associated with a higher MHQoL at baseline. The follow-up regression model revealed that a higher MHQoL at follow-up was significantly associated with less psychological distress at follow-up (β = −0.62, *p* < 0.01) and more weight loss at follow-up (β = 0.17, *p* < 0.05). Both the models were significant, accounting for 67.0% of the total baseline variance and 63.6% of the total follow-up variance (Table 4).

**Table 4 nutrients-16-01068-t004:** Multiple linear regression analysis: standardised regression coefficients (beta, β) for predictors of body mass index, EDE-Q global score, binge eating, SF12PCS, and SF12MCS at baseline (T0), and 12-month follow-up (T1) assessments.

	Baseline and Follow-Up Measures: Standardised Regression Coefficients (beta, β)									
Baseline and Follow-Up Measures #	Baseline EDE-QS	Follow-Up EDE-QS	Baseline Binge Eating	Follow-Up Binge Eating	Baseline K10	Follow-Up K10	Baseline PCS	Follow-Up PCS	Baseline MCS	Follow-Up MCS
	*β*	*β*	*Β*	*β*	*β*	*β*	*β*	*β*	*β*	*β*
Gender (female, male)	-	-	-	-	−0.10	−0.03		−0.08	0.06	0.05
Age (continuous)	−0.09	-	−0.20 *	-	−0.08	-	−0.33 **		−0.11	-
Aboriginal	-	0.04	-	-	0.17 *	0.01			−0.02	−0.02
Baseline BMI	-	-	-	-	-	-	−0.30 **			-
Baseline EDE-QS Global	-	0.23 **	-	-	0.58 **	NI			−0.01	NI
Baseline binge eating	-	-	-	0.23 **	-	-			0.17 **	-
Baseline K10	0.63 **	NI	0.42 **	NI	-	0.58 **			−0.82 **	NI
Baseline PCS		-	-	-		-		0.49 **	−0.30 **	-
Baseline MCS		-		-		-				0.15
Follow-up EDE-QS Global		-		-		0.31 **				−0.01
Follow-up binge eating		-		-		-				-
Follow-up K10		0.32 **		0.39 **		-		−0.13		−0.62 **
Follow-up PCS										-
Follow-up BMI										-
Follow-up weight loss		−0.35 **				−0.08				0.17 *
Adjusted R-square	0.425 **	0.417 *	0.256 **	0.237 **	0.462 **	0.583 **	0.147 **	0.294 **	0.670 **	0.636 **

Notes: # Measures found with significant correlations in Table 3 were included in multiple linear regression analysis. NI, not included in follow-up model. In Table 3, baseline measure was found significantly correlated with same measures at follow-up. To avoid multicollinearity, only significant follow-up measures were included in follow-up model. K10 = Kessler 10; EDE-QS = Eating Disorder Examination Questionnaire Short Version; PCS = Physical composite score; MCS = Mental composite score. * Significant at *p* < 0.05; ** significant at *p* < 0.01.

## 4. Discussion

While many WMPs focus on weight loss itself, research has suggested the importance of interventions that also address other factors, including co-morbidities and MHQoL [18,19]. This study demonstrates that patients with Class 3 obesity attending a multidisciplinary WMP for one year had poor mental health outcomes and quality of life at baseline, which were significantly improved at 12-month follow-up. With the exception of the improvements in eating disorder risk and mental health quality of life, improvements in all the other psychological outcomes were not associated with greater weight loss. The clinical and demographic features associated with improvements in each psychological outcome are addressed individually in turn.

The improvements in eating disorder risk were associated with a lower eating disorder risk at baseline, less psychological distress at follow-up, and more weight loss at follow-up. The relationship between psychological distress and eating disorder risk may be explained by a reduced need to cope with negative emotions through eating, and subsequently less-negative feelings about oneself as a result of emotional eating [64]. Given that individuals with obesity have a higher risk of developing eating disorders [65], and binge eating disorder frequently co-occurs with obesity [66], these results emphasise the importance of targeting psychological distress to improve the eating disorder outcomes in adults with obesity. The results also demonstrate that females had more psychological distress and a poorer MHQoL compared to the males in this cohort, which may suggest a gender-specific strategy to deal with these psychological outcomes. However, employment status did not demonstrate a statistical difference in the psychological outcome measures. The improvements in binge eating frequency were associated with a lower baseline binge eating frequency and less psychological distress at the follow-up. These results are in line with a previous trial that found significant improvements in both psychological distress and binge eating when managing obesity [67]. A study by Colles and colleagues [68] has also shown that the loss of control experienced during a binge-eating episode is central to the psychological distress experienced. Binge eating has been identified as one of the main forms of disordered eating in obesity [69], and binge eating disorder is one of the most prevalent psychiatric disorders in patients with obesity [70]. Our results suggest that treating psychological distress can help to reduce the binge eating frequency, which has been found to be distressing for individuals with obesity, independent of their actual weight [71].

Less psychological distress at baseline and a lower eating disorder risk at follow-up were associated with reduced psychological distress at follow-up. This is supported by previous evidence demonstrating that increased concern with eating, weight, and shape, as well as increased dietary restraint, all predicted the psychological distress [72] and quality of life [65] among women with obesity. The improvement in eating disorder risk following this WMP may operate via a cognitive shift in what individuals base their self-evaluation on (i.e., from weight and shape to other characteristics), or via a behavioural shift in reducing the frequency of disordered eating behaviours previously used as coping mechanisms. Future studies should delineate the specific mechanisms by which WMPs could improve the eating disorder risk and treat psychological distress.

The improvements in PHQoL were associated with a higher PHQoL at baseline, while an improved MHQoL was associated with less psychological distress and greater weight loss at follow-up. Interestingly, weight loss was not associated with improvements in PHQoL. This appears to contradict the previous findings that suggest improvements in PHQoL may be affected by weight reduction, where the participants had Class 1 obesity (BMI < 35 kg/m^2^) [36,38]. Our findings may be because most participants had Class 3 obesity (BMI > 40 kg/m^2^) at baseline, and perhaps the degree of weight loss may not have been substantial enough to produce significant improvements in PHQoL. Further, our study population had a greater rate of comorbidities compared to those of other studies, which may have impacted ratings of PHQoL. In support of this, other studies examining the QoL outcome following bariatric surgery have found improvements in PHQoL, which have been hypothesised as being due to weight loss and a reduction in medical comorbidities [73]. Importantly, the improvements in PHQoL were greater after surgical procedures that resulted in greater weight loss [73,74].

In addition, an improved MHQoL at follow-up was significantly correlated with a higher baseline MHQoL. However, only less psychological distress and higher weight loss remained statistically significant after adjustment. This suggests that the benefits of losing weight and reducing psychological distress outweigh the benefits of having a better initial MHQoL. Therefore, to obtain greater improvements in MHQoL after participation in a WMP, it would be worthwhile to focus on reducing psychological distress and increasing weight loss regardless of the initial MHQoL scores. Our study adds to the existing literature on the inconsistent relationship between MHQoL and weight loss [32,37,38]. Future studies should focus on identifying potential predictors that mediate this relationship and could explain the discrepancy in findings across studies.

### 4.1. Strengths and Limitations

Our findings add to the literature on the efficacy of weight management interventions by shedding light on which variables are linked to improvements in mental health outcomes and what interventions need to be targeted in addition to weight loss to benefit the individuals participating in a multidisciplinary WMP. Another key advantage of the current study was that it was conducted in a real-world clinical setting, which, therefore, makes the findings more generalisable to everyday patients in real-life clinical settings.

The main limitation in our study was that there was a significant number of dropouts at 12 months in this study. However, the main outcome measures at the baseline did not differ significantly between the ‘retained’ and ‘lost to follow-up’ groups, e.g., the mean BMI score (51.1 vs. 51.9; *p* = 0.496); mean K10 score 25.9 vs. 25.7; *p* = 0.890); mean EDE-QS global score (15.9 vs. 16.0; *p* = 0.979); mean binge eating episodes (0.90 vs. 1.00; *p* = 0.259); and mean MCS score (40.1 vs. 39.4; *p* = 0.616). A significantly higher proportion of men (23.0% vs. 35.6%; *p* = 0.0260) and younger (mean age: 51.3 vs. 46.9; *p* = 0.008) participants were lost to the 12-month post-assessment follow-up. These finding indicates that the dropouts did not have a major impact on the results, although caution is required when interpreting these results concerning the gender and age of the participants. Further, given that the program involved a pharmacological intervention, it cannot be determined whether the improvements observed were due to solely to behavioural aspects. Future studies could identify the unique contribution of behaviour modification components such as goal setting, self-monitoring, and cognitive restructuring [75,76]. Further research is also required to determine if this improvement in psychological outcomes following a WMP that focuses on treating psychological distress and reducing the eating disorder risk persists in the longer term.

Another limitation was the use of self-reported questionnaires to assess the risk of eating disorders, psychological distress, and quality of life. While these questionnaires have been validated and are psychometrically sound, they may have been subject to recall bias or reporting bias. Future studies could utilise interviews to assess the risk or make a clinical diagnosis. Furthermore, there are limitations to the use of the epidemiological tool of BMI that risks misclassifying obesity [39], particularly in people of diverse ethnicities [77]. Therefore, the findings of this study need to be replicated with more accurate tools to measure people’s metabolic status, such as DEXA scans [39,45], and for those with a higher weight beyond the maximum capacity of DEXA, circumference and bioimpedance.

### 4.2. Study Implications

The key finding was that among people with Class 3 obesity, weight loss alone did not predict the improvements in mental health outcomes, despite being the key measure of success in many WMPs. In accordance with the recent sentiment that the mental health outcomes of people with obesity are a critical aspect of weight management, our study demonstrates the importance of examining the psychological correlates of obesity and targeting these specifically in WMPs. While the different intervention components may have contributed to the improvements in mental health outcomes, the impact of the patients’ comfort within the WMP and the number of times they were seen by WMP staff members on the WMP outputs are not known. Future research should, therefore, focus on revising existing WMPs with the goal of producing improvements in these key psychological correlates in conjunction with the focus on improvement in physical/metabolic health and dissecting out which component of the WMP contributed most to the improvement in the outcomes.

Another key finding was that MHQoL was one of two outcomes that were associated with weight loss, contributing to previous research on this inconsistent relationship. On the other hand, PHQoL was not associated with weight loss, contrary to the previous findings. Further research is needed on both these aspects of health related QoL in the context of Class 3 obesity to inform how it can be addressed in WMPs.

## 5. Conclusions

The psychological aspects of obesity are critical to address in WMPs, given that many individuals with obesity have poorer mental health outcomes. Our study has identified that multidisciplinary WMPs can result in improvements in the mental health outcomes. Our intervention was associated with reduced self-reported eating disorder symptoms, reduced psychological distress, and improved mental health quality of life. The improvements in most mental health outcomes, however, were not predicted by weight loss alone. Reduced psychological distress was associated with reduced binge eating and a lower eating disorder risk with a lower baseline eating disorder risk at follow-up, less psychological distress, and greater weight loss. These findings support the inclusion of targeted psychological interventions in addition to behavioural WMPs to achieve improved outcomes among people with Class 3 obesity. Given that most WMPs focus on weight loss, our study provides evidence that there are wider benefits for patients in terms of mental health outcomes and quality of life that may be enhanced by targeted psychological interventions. Future studies should focus on replicating this study with an appropriate evaluation and tools to measure body composition and identify the specific mechanisms that can contribute to improvements, and as well as modify the WMP components to maximise the improvements in psychological distress and eating disorder risk.

## Figures and Tables

**Table 1 nutrients-16-01068-t001:** Baseline (T0) characteristics of the 115 participants at both baseline and 12-month follow-up (T1) assessments.

	*n*	%
#All	115	100.0
Gender		
Female	88	76.5
Male	27	23.5
Age groups (in years)		
18–29	9	7.8
30–44	29	25.2
45–64	55	47.8
65 and above	22	19.1
Ethnicity		
Caucasian	90	78.3
Aboriginal and Torres State Islander	9	7.8
Other	16	13.9
Employment Status		
Fulltime employed	19	16.7
Part time employed	12	10.5
Pension/Retired	54	47.4
Unemployed/other	29	25.4
Binge eating at baseline (days per week)		
0 Days	52	46.0
1–2 days	36	31.9
3–5 days	12	10.6
6–7 days	13	11.5
Hypertension	78	67.8
Fatty liver diseases	30	26.1
Gastro oesophageal reflux disorder	48	42.1
Type 2 diabetes	65	56.5
	*n*	Mean (SD)
Age/years	115	51.3 (13.8)
Cholesterol (mmol/L)	103	4.5 (1.1)
Triglyceride (mmol/L)	101	1.9 (1.0)
HDL (mmol/L)	76	1.2 (0.3)
LDL (mmol/L)	74	2.5 (1.0)
Physical function	112	32.6 (25.6)
Baseline body mass index (BMI)	115	51.1 (8.6)
Baseline psychological distress (K10)	113	25.9 (9.8)
Baseline EDE-QS Global	114	15.9 (6.7)
Baseline binge eating	113	0.9 (1.0)
Baseline physical quality of life (PCS)	112	29.5 (10.2)
Baseline mental quality of life (MCS)	112	40.1 (12.4)
Baseline body weight	115	140.6 (26.0)

Notes: K10 = Kessler 10; EDE-QS = Eating Disorder Examination Questionnaire Short Version; PCS = Physical composite score; MCS = Mental composite score; LDL = Low-density lipoprotein; HDL = High-density lipoprotein. # Due to exclusion of missing and not stated cases, total number may not always add up to 115.

**Table 2 nutrients-16-01068-t002:** Mean total score with standard deviation (SD), correlation co-efficient, and effect size estimates (Cohen’s *d* with 95% CI) for outcome measures from pairwise matched sample: Baseline (T0) and 12-month follow-up (T1) for women, men, and all participants, respectively.

Measures/Pairwise Matched Females, Males, and Total Sample	Gender	*n*	Baseline: Mean (SD)	12-Month Follow-Up Mean (SD)	T0 vs. T1: *p*-Values: Paired *t*-Test	Correlation: T0 and T1 (r)	Effect Size: Cohens’ *d* (95%CI)
Body mass index	Female	88	52.1 (8.6)	49.0 (9,0)	<0.001	0.91 **	0.82 (0.58, 1.06)
	Male	27	48.0 (8.1)	44.8 (7.1)	<0.001	0.93 **	1.09 (0.60, 1.56)
	Total	115	51.1 (8.6)	48.0 (8.7)	<0.001	0.91 **	0.87 (0.65, 1.08)
	*p*-Values: *t*-test		0.014	0.007			
Body weight	Female	88	138.1 (26.1)	130.1 (27.0)	<0.001	0.93 **	0.83 (0.59, 1.07)
	Male	27	148.8 (24.6)	138.5 (23.0)	<0.001	0.94 **	1.22 (0.72, 1.72)
	Total	115	140.6 (26.0)	132.0 (26.2)	<0.001	0.94 **	0.91 (0.69, 1.13)
	*p*-Values: *t*-test		0.029	0.058			
Psychological distress	Female	85	27.0 (9.9)	21.8 (9.4)	<0.001	0.67 **	0.68 (0.44, 0.91)
	Male	27	22.1 (9.0)	17.6 (7.7)	0.001	0.66 **	0.65 (0.22, 1.06)
	Total	112	25.8 (9.8)	20.8 (9.1)	<0.001	0.69 **	0.67 (0.46, 0.87)
	*p*-Values: *t*-test		0.009	0.009			
EDE-QS	Female	85	16.4 (6.7)	13.8 (6.3)	0.001	0.37 **	0.36 (014, 0.58)
	Male	27	14.4 (7.0)	12.2 (6.3)	0.046	0.52 **	0.34 (−0.06, 0.72)
	Total	112	15.9 (6.7)	13.4(6.3)	0.000	0.42 **	0.35 (0.16, 0.55)
	*p*-Values: *t*-test		0.093	0.120			
Binge eating	Female	83	0.9 (1.0)	0.7 (0.8)	0.019	0.33 **	0.23 (0.01, 0.45)
	Male	25	0.8 (1.0)	0.5 (0.6)	0.083	0.28	0.29 (−0.12, 0.68)
	Total	108	0.9 (1.0)	0.6 (0.7)	0.006	0.33 **	0.24 (0.05, 0,43)
	*p*-Values: *t*-test		0.148	0.108			
Physical quality of life	Female	85	29.9 (10.5)	37.0 (11.4)	<0.001	0.55 **	−0.69 (−0.92, −0.43)
	Male	27	28.3 (9.2)	34.7 (9.6)	0.001	0.50 **	−0.67 (−1.09, −0.25)
	Total	112	29.5 (10.2)	36.5 (11.0)	<0.001	0.54 **	−0.69 (−0.89, −0.48)
	*p*-Values: *t*-test		0.227	0.162			
Mental quality of life	Female	85	38.5 (11.9)	42.6 (13.3)	0.003	0.49 **	−0.32 (−0.54, −0.10)
	Male	27	45.2 (12.9)	49.8 (11.7)	0.002	0.82 **	−0.61 (−1.02, −0.19)
	Total	112	40.1 (12.4)	44.3 (13.2)	<0.001	0.59 **	−0.36 (−0.55, −0.17)
	*p*-Values: *t*-test		0.010	0.005			

Notes Cohen’s *d* indicates small effect = 0.20; medium effect = 0.50; large effect = 0.80. EDE-QS = Eating Disorder Examination Questionnaire Short Version. ** Correlation coefficients are significant at *p* < 0.01.

**Table 3 nutrients-16-01068-t003:** Correlation among baseline and follow-up measures.

	Measures and Correlation Coefficient (r)												
Measures	Baseline BMI	Baseline EDE-QS	Baseline Binge Eating	Baseline K10	Baseline PCS	Baseline MCS	Follow-Up BMI	Follow-Up EDEQS	Follow-Up Binge Eating	Follow-Up K10	Follow-Up PCS	Follow-Up MCS	Follow-Up Weight Loss
Gender (female/male)	−0.20 *	−0.13	−0.10	−0.22 *	−0.07	0.23 *	−0.21 *	−0.11	−0.11	−0.20 *	−0.09	0.23 *	
Age (continuous)	−0.18	−0.27 **	−0.32 **	−0.30 **	−0.28 **	0.24 *	−0.26 **	−0.08	−0.10	−0.14	−0.21 *	0.11	
Aboriginal	0.24 *	0.18	0.17	0.32 **	0.06	−0.28 **	0.28 **	0.21 *	0.16	0.28 **	0.13	−0.26 *	
Baseline BMI	-	0.11	0.05	0.16	−0.24 *	−0.07	0.91 **	0.09	0.04	0.15	−0.05	−0.04	
Baseline EDE-QS Global	0.11	-	-	0.65 **	−0.04	−0.50 **	0.15	0.42 **	-	0.49 **	−0.02	−0.38 **	
Baseline binge eating	0.05	-	-	0.48 **	−0.01	−0.23 *	0.04	-	0.33 **	0.27 **	0.12	−0.21 *	
Baseline K10	0.16	0.65 **	0.48 **	-	−0.10	−0.78 **	0.20 *	0.33 **	0.42 **	0.69 **	−0.10	−0.55 **	
Baseline PCS	−0.24 *	−0.04	−0.01	−0.10	-	−0.19 *	−0.21 *	−0.16	−0.03	−0.18	0.54 **	0.05	
Baseline MCS	−0.07	−0.50 **	−0.23 *	−0.78 **	−0.19 *	-	−0.14	−0.28 **	−0.28 **	−0.62 **	0.07	0.59 **	
Follow-up BMI							-	0.28 **	0.10	0.27 **	−0.12	−0.20 *	
Follow-up EDE-QS Global							0.28 **	-	-	0.55 **	−0.15	−0.48 **	−0.45 **
Follow-up binge eating							0.10	-	-	0.45 **	−0.02	−0.43 **	−0.14
Follow-up K10							0.27 **	0.55 **	0.45 **	-	−0.20 *	−0.78 **	−0.28 **
Follow-up PCS							−0.12	−0.15	−0.02	−0.20 *	-	0.06	0.14
Follow-up MCS							−0.20 *	−0.48 **	−0.43 **	−0.78 **	0.06	-	0.37 **
Follow-up weight loss							-	−0.45 **	−0.14	−0.28 **	0.14	0.37 **	-

* Significant at *p* < 0.05; ** significant at *p* < 0.01. BMI = body mass index; K10 = Kessler 10; EDE-QS = Eating Disorder Examination Questionnaire Short Version; PCS = Physical composite score; MCS = Mental composite score.

## Data Availability

The data used to support the findings of this study are available from the corresponding author upon request as these are hospital data.

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
