# Peer review of "Predictors of Mental Health Outcomes in a Multidisciplinary Weight Management Program for Class 3 Obesity"

_nutrients, 2024, doi:10.3390/nu16071068_

Round 1

Reviewer 1 Report

Comments and Suggestions for Authors

First of all, thank you for the opportunity to review this article. The article “Predictors of Mental Health Outcomes in a Multidisciplinary 2 Weight Management Program for Class 3 Obesity” aims to know, through a retrospective study, the predictive factors of improvement in mental health outcomes in a cohort of patients with class 3 obesity.

The article follows the structure proposed for a research study. The title and abstract summarize the content of the article, as do the keywords. However, in the abstract it is indicated that it is a retrospective study and in the Materials and Methods section, it is indicated that a prospective cohort was used. This point should be clarified.

Regarding bibliographic references, it must be taken into account that there is a lack of access date (for example, in reference 1), and there are references to very old publications, for example reference 2 from 1999. Although, they are references important in the categorization of obesity, I consider that it should be qualified. On the other hand, reference 65 could be complemented with more recent studies.

Regarding the introduction, it is clear and allows us to define the situation of obesity and the relationship between obesity and adverse mental health issues.

The methodology is adequate, indicating each of the questionnaires used, as well as the collection and analysis of the data. It is not indicated whether participants signed an informed consent.

Regarding the results and discussion, they are clear. The results should indicate the dropout rate of participants who initially started the program. On the other hand, in the discussion, I believe that an attempt should be made to discuss, if possible, aspects such as gender and employment status with the results of the study. I think that the great statistical analysis carried out could be accompanied by broader discussion

Thank you, in the limitations of the study, for raising the difficulty of being able to establish the limitation regarding whether the improvements that have appeared can be affected by the pharmacological treatment.

The conclusion responds to the objective and approach of the study.

Author Response

27th March 2024

Re: Manuscript Number: nutrients-2919381

Response to Reviewer 1:

First of all, we thank you for the opportunity to review this article. The article “Predictors of Mental Health Outcomes in a Multidisciplinary 2 Weight Management Program for Class 3 Obesity” aims to know, through a retrospective study, the predictive factors of improvement in mental health outcomes in a cohort of patients with class 3 obesity.

The article follows the structure proposed for a research study. The title and abstract summarize the content of the article, as do the keywords.

Thank you for your feedback on our paper.

Comment 1: However, in the abstract it is indicated that it is a retrospective study and, in the Materials, and Methods section, it is indicated that a prospective cohort was used. This point should be clarified.

Author Response: Thank you for highlighting this. We have now revised the Material and Methods section to state that this was a retrospective study.

Comment 2: Regarding bibliographic references, it must be taken into account that there is a lack of access date (for example, in reference 1), and there are references to very old publications, for example reference 2 from 1999. Although, they are references important in the categorization of obesity, I consider that it should be qualified. On the other hand, reference 65 could be complemented with more recent studies.

Author Response: We have populated the date of access for reference 1. Reference 2 was cited alongside reference 3 that was published in 2023. A more updated reference 66 has been added to complement reference 65.

Comment 3: Regarding the introduction, it is clear and allows us to define the situation of obesity and the relationship between obesity and adverse mental health issues.

Author Response Thank you for your feedback.

Comment 4: The methodology is adequate, indicating each of the questionnaires used, as well as the collection and analysis of the data. It is not indicated whether participants signed an informed consent.

Author response: The following informed consent statement has been included in section 2.2. of the methods section –

The data was anonymised data that was collected routinely in the clinic and was granted a waiver to signed informed consent as part of a quality improvement project (Reference: CT_08_2022).

Comment 5: Regarding the results and discussion, they are clear. The results should indicate the dropout rate of participants who initially started the program.

Author Response: Thank you for pointing this out. We have now included details about dropout rates (62.4%) in the last paragraph of the section ‘2.1. Study Design and Participants.

 “During the study period a total of 306 eligible participants were enrolled in the WMP. Prior to program intervention all enrolled participants were assessed through a self-repot baseline questionnaire and participants were re-assessed at 12-months post-assessment follow-up. Of the 306 enrolled participants only 115 (37.6%) completed both baseline and 12-month post follow-up assessment and the remaining 191 (62.4%) did not complete the 12-month post-assessment follow-up. The 115 participants those who completed assessment at both time points are included in this study.”

As the drop-out rate is quite high (62.4%), we have examined the differences of main outcome measures between the dropout participants and those who retained in the study. Accordingly, the text in discussion section related to limitation are also revised.

“There was a significant drop-out at 12 months in this study. However, the main outcome measures at baseline did not differ significantly between the ‘retained’ and ‘lost to follow-up’ groups e.g., mean BMI score (51.1 vs. 51.9; P= 0.496); mean K10 score 25.9 vs. 25.7; P= 0.890); mean EDEQ Global score (15.9 vs. 16.0; P= 0.979); mean binge eating episodes (0.90 vs. 1.00; P= 0.259); and mean MCS score (40.1 vs. 39.4; P= 0.616). Significantly higher proportion of men (23.0% vs. 35.6%; P= 0.0260) and younger (mean age: 51.3 vs. 46.9; p=0.008) participants were lost to 12-month post-assessment follow-up. These finding indicates that the drop out did not have a major impact on the results, although caution is required when interpreting these results concerning gender and age of participants.”

Comment 6: On the other hand, in the discussion, I believe that an attempt should be made to discuss, if possible, aspects such as gender and employment status with the results of the study. I think that the great statistical analysis carried out could be accompanied by broader discussion.

Author Response: We thank the authors for highlighting gender and employment status. As discussed in the results section and Table 3, being female was significantly associated with higher BMI, higher psychological distress and lower MHQoL. This has now been highlighted in the discussion. The initial analysis found that there was no statistical difference in psychological outcome measures when comparing the different employment status groups (see the Tables below). Also sample sizes for different employment groups was not statistically sufficient to explore the effect size for each of the employment groups respectively and thus employment status has been excluded from the statistical analyses. In the revised version this has also been mentioned in the methods and discussion section.

Table A. Mean score for main outcome measures at baseline by employment status

Main outcome measures at baseline

Number

Baseline BMI

Baseline EDEQS

Baseline binge eating

Baseline K10

Baseline MCS

Employment status

Mean

Mean

Mean

Mean

Mean

Employed full time

19

50.5

16.2

0.9

23.4

44.7

Employed part time

12

51.2

14.3

1.1

22.6

45.9

Pension/Retired

54

52.1

16.1

0.8

26.9

38.4

Unemployed others

29

49.4

16.0

0.9

26.9

37.7

Total

114

51.0

15.9

0.9

25.9

40.1

P-values from F-test

p=0.59

p=0.86

p=0.81

p=0.34

p=0.06

Table B. Mean score for main outcome measures at follow-up by employment status

Main outcome measures at follow-up

Follow-up BMI

Follow-up EDEQS

Follow-up binge eating

Follow-up K10

Follow-up MCS

Follow-up weight loss

Employment status

Number

Mean

Mean

Mean

Mean

Mean

Mean

Employed full time

19

47.9

13.4

0.5

18.2

48.7

7.3

Employed part time

12

47.1

11.9

0.6

17.5

49.5

12.0

Pension/Retired

54

48.9

14.2

0.7

22.6

42.1

8.5

Unemployed others

29

46.4

12.6

0.6

20.9

43.6

8.1

Total

114

47.9

13.4

0.6

20.9

44.3

8.5

P-values from F-test

p=0.66

p=0.59

p=0.81

p=0.16

p=0.13

p=0.57

Comment 7:  Thank you, in the limitations of the study, for raising the difficulty of being able to establish the limitation regarding whether the improvements that have appeared can be affected by the pharmacological treatment.

Author Response: Thank you for this feedback.

Comment 8: The conclusion responds to the objective and approach of the study.

Author Response: Thank you for this feedback.

Reviewer 2 Report

Comments and Suggestions for Authors

Reviewer's Comments to Authors:

Recommendation: Major Revision

Manuscript Nutrients-2919381, titled "Predictors of Mental Health Outcomes in a Multidisciplinary Weight Management Program for Class 3 Obesity," presents an intriguing exploration of obesity as a chronic disease, its psychological correlates, and its association with eating disorders and mental health. Overall, the manuscript is well-prepared, but there are several areas that require attention:

ABSTRACT:

Clarify the purpose of the research to provide a more precise understanding for readers.

Revise the presentation of findings to enhance clarity and highlight the originality of the research.

INTRODUCTION:

Acknowledge that BMI is not the sole measure of obesity. Consider discussing potential limitations and alternative measures, particularly concerning different ethnic groups.

Clearly articulate the aim of the study, especially concerning the evaluation of underlying mechanisms between BMI and other clinical and sociodemographic measures. Consider addressing potential confounding factors such as comorbidities.

MATERIALS AND METHODS:

Specify whether weight and height were measured directly or self-reported by patients. Provide clarity on data collection methods.

Include comprehensive data on physical activity levels, comorbidities, and other relevant variables to ensure a thorough understanding of the study population.

RESULTS and DISCUSSION:

Consider consolidating tables or presenting them more coherently to enhance readability. If necessary, provide supplementary material as promised.

Strengthen the discussion by addressing any missing considerations highlighted in the introduction. Emphasize key findings and their implications for the research field.

CONCLUSION:

Develop a clear and concise conclusion summarizing the main findings and their significance. Reiterate the originality of the research and its scientific justification.

Ensure that the most important results are adequately emphasized to underscore their significance to the field of study.

Addressing these points will enhance the clarity, completeness, and impact of the manuscript, thereby strengthening its contribution to the existing literature on obesity and mental health outcomes.

Author Response

27th March 2024
Re: Manuscript Number: nutrients-2919381

Response to Reviewer 2: 
Manuscript Nutrients-2919381, titled "Predictors of Mental Health Outcomes in a Multidisciplinary Weight Management Program for Class 3 Obesity," presents an intriguing exploration of obesity as a chronic disease, its psychological correlates, and its association with eating disorders and mental health. Overall, the manuscript is well-prepared, but there are several areas that require attention:

Thank you for your feedback on our paper.

Comment 1: ABSTRACT:
Clarify the purpose of the research to provide a more precise understanding for readers.
Revise the presentation of findings to enhance clarity and highlight the originality of the research.

Author Response: The following has been added to the extract to highlight the originality of the research: Notably, eating disorder risk was associated with lower eating disorder risk at baseline, lower psychological distress at follow-up, and greater weight loss at follow-up. Our results also found an association between reduced psychological distress and reduced binge eating frequency. These findings support the inclusion components of obesity interventions that target the psychological correlates of obesity to support improved outcomes in people with class 3 obesity.

INTRODUCTION:
Comment 2: Acknowledge that BMI is not the sole measure of obesity. Consider discussing potential limitations and alternative measures, particularly concerning different ethnic groups. Furthermore, there are limitations in the use of BMI as an indicator of health risk, particularly in people of diverse ethnicities. 

Author Response: Thank you for this point, we have added a sentence in the limitations of the study highlighting this important point about BMI. And added a reference for this.

Comment 3: Clearly articulate the aim of the study, especially concerning the evaluation of underlying mechanisms between BMI and other clinical and sociodemographic measures. Consider addressing potential confounding factors such as comorbidities.
Author Response: Thank you for highlighting this important point. We have now included these points in the last paragraph of the introduction.

MATERIALS AND METHODS:
Comment 4: Specify whether weight and height were measured directly or self-reported by patients. Provide clarity on data collection methods.

Author Response: Data on weight and height, medical comorbidities, current medications, and blood tests were directly collected during presentations to the clinic. Data on three questionnaires was done by the participants in the waiting room of the clinic: Eating Disorder Examination Questionnaire Short (EDE-QS), Kessler-10 Psychological Distress Scale (K10), and 36-Item Short Form Survey (SF-36). These are clearly indicated in ‘Data collection and Management’ section in the revised manuscript.

Comment 5: Include comprehensive data on physical activity levels, comorbidities, and other relevant variables to ensure a thorough understanding of the study population.

Author Response: As per suggestions, in Table 1 we added baseline data physical function, comorbidities and relevant measures. 

RESULTS and DISCUSSION
Comment 6: Consider consolidating tables or presenting them more coherently to enhance readability. If necessary, provide supplementary material as promised.

Author Response: We have consolidated, and re-formatted tables as recommended.

Comment 7: Strengthen the discussion by addressing any missing considerations highlighted in the introduction. Emphasize key findings and their implications for the research field.
Author Response: We thank the reviewer for this suggestion. We have now included the effect of gender and of employment status in the discussion, which was also suggested by the other reviewer.

CONCLUSION:
Comment 8: Develop a clear and concise conclusion summarizing the main findings and their significance. Reiterate the originality of the research and its scientific justification. Ensure that the most important results are adequately emphasized to underscore their significance to the field of study.

Author Response: Thank you for this feedback. We have made the following additions/amendments to the conclusion of this paper.
Psychological aspects of obesity are critical to address in WMPs, given that many individuals with obesity have poorer mental health outcomes. Our study has identified that multidisciplinary WMPs can result in improvements in mental health outcomes. In particular our intervention was associated with reduced self-reported eating disorder symptoms reduced psychological distress and improved mental health quality of life. Improvements in most mental health outcomes, however, were not predicted by weight loss alone. Reduced psychological distress was associated with reduced binge eating and lower eating disorder risk with lower baseline eating disorder risk, at follow up, lower psychological distress and greater weight loss. These findings support the inclusion of targeted psychological interventions in addition to behavioural WMPs in improved outcomes in people with class 3 obesity. Given that most WMPs focus on weight loss, our study provides evidence that there are wider benefits for the patient in terms of mental health outcomes and quality of life that may be enhanced by targeted psychological interventions. Future studies should focus on identifying the specific mechanisms that can contribute to improvements, and as well as modifying WMP components to maximise the improvements in psychological distress and eating disorder risk. 

Comment 9: Addressing these points will enhance the clarity, completeness, and impact of the manuscript, thereby strengthening its contribution to the existing literature on obesity and mental health outcomes.
Author Response: Thank you for your time to review our article. We have addressed all the issues and believe that your review has strengthened our manuscript.  

Round 2

Reviewer 2 Report

Comments and Suggestions for Authors

The work meets the requirements of the review and can be accepted for publication in its current form.

Author Response

Author Response:

Thank you for your comments.